# Assessment of Headache Characteristics, Impact, and Managing Techniques among Pharmacy and Nursing Undergraduates—An Observational Study

**DOI:** 10.3390/medicina59010130

**Published:** 2023-01-09

**Authors:** Adel S. Bashatah, Wajid Syed, Mahmood Basil A. Al-Rawi, Mohamed N. Al Arifi

**Affiliations:** 1Department of Nursing Administration & Education, College of Nursing, King Saud University, Riyadh 11451, Saudi Arabia; 2Department of Clinical Pharmacy, College of Pharmacy, King Saud University, Riyadh 11451, Saudi Arabia; 3Department of Optometry, College of Applied Medical Sciences, King Saud University, Riyadh 11451, Saudi Arabia

**Keywords:** headache, photophobia, phonophobia, medication therapies, academic performance

## Abstract

*Background and Objectives:* Many different forms of headaches can change or impact daily activity and quality of life, which increases the financial burden on society over time. Undergraduates who get headaches may be absent from attending lectures, perform less well on their daily tasks and academic achievement, as well as struggle to build and maintain relationships with peers and mentors. Therefore, this study aimed to assess the headache-related characteristics and managing approaches among Saudi pharmacy and nursing students at a Saudi university, in Riyadh, Saudi Arabia. *Materials and Methods:* A survey questionnaire was administered in this cross-sectional study to participants at a Saudi university, in Riyadh, Saudi Arabia. Participants included males. The sample size was calculated with Raosoft^®^ software. Data analysis was executed using IBM Statistic SPSS, and the level of statistical significance was set at *p* < 0.05. *Results:* A total of 236 participants completed the questionnaires. The majority, i.e., 218 (92.4%) of them, were male; in addition, 124 (52.5%) were aged between 26 and 30, 124 (52.5%) were pharmacy students, 112 (47.5%) were nursing students, and 134 (56.8%) were smokers. When asked about ever having at least one episode of headache during the week, 66.1% (*n* = 156) agreed that they had one episode of headache, although 57 (24.2%) of the students had a headache for five days during a week. With regard to the impact of headaches on everyday activities, only 34.7% of the students said that headache disrupted their regular activities. Almost 41% of the students agreed that headache impacted their academic performance. Nearly 34% of students (*n* = 79) who had headaches considered napping, while 33% (*n* = 64) took painkillers and anti-inflammatory medicines, and 25% (*n* = 59) considered taking caffeine. In this study, the participants’ ages and headache severity were strongly associated (*p* = 0.0001). More pharmacy students (66.1%) reported having severe headaches than nursing students (14.3%) (*p* = 0.0001). *Conclusions:* The current findings revealed that most of the undergraduates suffered from headaches, and the intensity of the pain was moderate; furthermore, one in four undergraduates reported that headaches impacted their academic performance. Caffeine and simple analgesics and anti-inflammatories were used for headache relief.

## 1. Introduction

Headaches are a regular phenomenon in the daily life of most people, especially university students [1,2]. According to estimates, headaches are the second most frequent cause of disability-related concerns [1,2]. A headache may be accompanied by throbbing, ongoing, mild, or intense discomfort in the head or face [2]. However, there are effective headache therapies available, such as medications, stress reduction techniques, and biofeedback [3]. Primary and secondary headaches are the two categories that the International Classification of Headache Disorders (ICHD-III) uses to categorize headaches [4]. A primary headache has no known cause, while a secondary headache is brought on by another ailment that causes inflammation of pain-sensitive tissues. Different types of headaches could be brought on by a vascular illness, a brain tumor, a systemic infection, or a head injury [5,6].

Regardless of gender or demographics, headaches are extremely challenging and can be incredibly interfaced. Aside from its issues, headaches are becoming more common in both adults and university students [7,8,9]. It has been established that headache-related issues affect a patient’s daily activities, education, and relationships, especially among undergraduates [10,11], as well as their treatment costs and clinic visits. According to the Global Burden of Disease (GBD) 2013 report, migraine was the sixth-highest cause of years lost to disability (YLD), whereas headache disorders were rated third [5,7].

In the United States of America, there are estimated to be 5 million individuals who get headaches [12]. About 30 to 40 million adult Americans suffer from migraine, the most well-known type of headache condition [10,11]. According to estimates, 15% of adults in the United Kingdom suffer from headaches, with women outnumbering males three to one [12]. According to a population-based national assessment of the burden of headaches in Ethiopia, the three most common types of headaches were described as migraine (17.7%), tension-type headache (TTH) (20.6%), migraine probably caused by taking too many medications, and headache yesterday (6.4%) [13]. On the other hand, it was revealed that headache incidence and prevalence were higher among undergraduates. For instance, a prior study by Birru et al. found that migraines and tension-type headaches were the two most common types of headaches, accounting for 81.1% [14]. In Saudi Arabia, Almesned et al. found a prevalence of 53.78% of headaches among medical students [15]. The prevalence was higher among pharmacy students, at 88.7% [16]. However, researchers at King Faisal University found a prevalence of 41.6% of headaches, and migraine prevalence was 58.4% [17]. According to a recent survey among female students, there was a prevalence of 68.4% of headaches [18].

In GBD 2016, migraine was the first most prevalent disabling disease among the 15–49 age group [19]. According to the Global Burden of Disease 2019 study, migraine was the second leading cause of disability and the leading cause among young women aged 50 years. Every year, GBD estimates are updated to track the burden of disease around the world, thereby anticipating future healthcare demands [20]. There has been extensive research conducted in the past decade on headaches among children, adolescents, students, and adults throughout the world to examine their prevalence, characteristics, and management strategies [12,13,19,20,21,22].

Healthcare education has evolved and developed, and there is a prospect of a better healthcare system. Pharmacy and nursing students are one of the most valuable healthcare professionals, responsible for safely and quickly diagnosing and managing headaches. These data can be used to assess whether or not students believe that parts of their education (e.g., the amount of coursework) cause their headaches, as well as how much they believe headaches negatively affect their academic performance (e.g., grades). Therefore, there is a need to investigate the characteristics and management of headache pain between pharmacy and nursing students in Saudi Arabia. The purpose of this study was to describe the approaches and management strategies for headaches between Saudi pharmacy and nursing students.

## 2. Methods

A cross-sectional prospective study was conducted in a Saudi university in Riyadh Saudi Arabia over 3 months in 2022 to assess the prevalence and characteristics of headaches among nursing and pharmacy undergraduates. Saudi nationals, students living in the university hotel, those pursuing the third and fourth year of their courses and willing to provide informed consent, students who suffered from headaches, and those willing to complete the questionnaires were included, while students from other courses and levels were excluded. We opted for a convincing sampling procedure to collect the data from the targeted population.

### 2.1. Sample Size Estimation

There were approximately 500 undergraduate students enrolled in PharmD and Nursing courses. Similar to many previous studies [23,24,25,26,27,28,29,30,31,32], we estimated the required sample size with a 95% CI and a 5% margin of error using an online calculator (http://www.raosoft.com/samplesize.html, last accessed on 6 January 2023). We assumed that the response distribution for each question would be 50% because we were uninformed of the probable outcomes for each question. Despite the fact that the sample size was supposed to be 218, we decided to poll at least 300 students to ensure more reliability.

The questionnaires for this study were adopted from previous studies [15,16,23]. The study questionnaires were divided into three sections. The Section 1 collected demographic information of the undergraduates, which include age, gender, course of study, smoking status, and the presence of any chronic disease. The second part of the study collected information about the frequency and characteristics of the headache and management approaches, with a total of 9 items. The last part of the study collected information on the impact headaches can have on daily activity and academic performance and the sensitivity of headaches, with a total of 6 items. All these questionnaires were assessed on a multiple-choice and binary scale. The pain intensity was calculated by assuming that 0 means no pain; 1–3 indicates mild pain; 4–7 indicates moderate pain; and 8–10 indicates severe pain on the rating scale of 0–10.

The designed questionnaire was translated into Arabic local language with the help of a senior professor who is a native of Saudi Arabia and an expert in translating the questionnaire, using forward and backward translation procedures. The final questionnaires were validated in two steps. First, the initial draft of the questionnaires was evaluated by a research expert in the related field, to check the accuracy of the content and flow of the questionnaire. Secondly, a pilot study was conducted among a randomly selected sample of (*n* = 30) undergraduates to give their opinions. Any amendments from the pilot study were then unified into the final questionnaire. The reliability test was performed by calculating Cronbach’s alpha using SPSS v.26, which was found to be 0.85. The data from the pilot study were not included in the final analysis. The final questionnaire was then distributed using an online survey portal.

The data collection was carried out by using a convenience sampling procedure from the targeted population; data collection initially began from the college of pharmacy, and later, the data were collected from the college of nursing. For data collection, a researcher was appointed, who made continuous follow-ups and forwarded reminders to complete the study. We designed an electronic survey link to collect the data, using google forms, and the designed link was distributed to the targeted population using a social media platform (WhatsApp). The data collection was carried out until the required sample size was obtained.

### 2.2. Data Analysis

Initially, the data were entered into MS Excel and then analyzed using the Statistical Package for the Social Sciences (SPSS) (version 26 for Windows (SPSS Inc., Chicago, IL, USA). Descriptive statistics, e.g., frequencies and percentages, were used to summarize the data. Mean and standard deviation were also calculated for some of the variables. The association between the dependent and independent variables was collected using a chi-square test, and a *p*-value of <0.005 was considered a statistically significant difference.

## 3. Results

A total of 236 participants completed the questionnaires. Of those, 218 (92.4%) were male, 124 (52.5%) were aged between 26 and 30, 124 (52.5%) were pharmacy students, and 112 (47.5 %) were nursing students. There were no instances of chronic disease among the majority of the undergraduates (91.7%), while 134 (56.8%) were smokers. The detailed data on the demographic characteristics of the respondents are shown in Table 1. 

When asked about ever having at least one episode of headache during the week, 66.1% (*n* = 156) agreed that they had one episode of headache, although 57 (24.2%) of the students had a headache for 5 days during a week. Concerning the previous question, regarding the number of episodes of headaches, 42.8% of the students had a headache for four days. More than one-third of the students reported speech disturbances, while 30.5% of them had a visual aura, with wavy lines or blind spots as reversible aura symptoms for headaches. Almost half (50.4%) of the undergraduates reported having headaches that were usually unilateral, while two-thirds of them had throbbing/pulsating type of headaches. Detailed information about the characteristics of the headache is given in Table 2.

With regard to the type of headache, findings revealed that frontal headaches were the most common, with 57 participants (24.2%), followed by generalized headaches at 23.3% (*n* = 55), temporal at 18.2% (*n* = 43), and occipital at 14.4% (*n* = 34) (Figure 1).

With regard to effects on everyday activities, only 34.7% of the students said that headache disrupted their regular activities, while 60.6% reported that headaches did not keep them from pursuing their routine activities. In addition, 32.6% said that headaches were usually associated with photophobia, although the majority, i.e., 76.7% (*n* = 181) of the students, also asserted that headaches were not usually associated with phonophobia. Only 25.8% of the students had a family history of headaches. Almost 40.7% of the students said that headaches impacted their academic performance (Table 3).

Although many students (76.3%) did not seek medical attention for headache management, 23.7% of them did. Nearly 34% of the students (*n* = 79) who had headaches considered napping, while 33% (*n* = 64) took painkillers and anti-inflammatory medicines, and 25% (*n* = 59) considered taking caffeine. In addition, the majority 65.3% (*n* = 154) of the students agreed that headaches stopped when they took medicine. Table 4 describes the frequency of headaches in respondents and their different management approaches for headaches.

In this study, the participants’ ages and headache severity were strongly associated. For example, 66.9% of participants between the ages of 26 and 30 reported having a severe headache, compared with other age groups (*p* = 0.0001). Similar to this, more pharmacy students (66.1%) than nursing students (14.3%) reported having severe headaches, indicating a significant difference between the two groups (*p* = 0.0001). Additionally, there was an association between the student’s smoking status, type of headache, and intensity (*p* = 0.0001). Additionally, Table 5 provides a thorough explanation of the relationship between participant demographics and headache intensity. As demonstrated in Table 6, there was also a substantial association between daily activity and headache intensity, although there was no association between headache intensity and academic activity.

## 4. Discussion

This study was used to evaluate characteristics and managing approaches towards headaches between undergraduate nursing and pharmacy students at King Saud University. Moreover, pharmacy and nursing students themselves are likely to encounter many episodes of headaches and may look for strategies to overcome headaches during their studies. There is a high prevalence of headaches in the general population and among university students based on our literature review and collected data. Numerous population-based studies have been conducted on headaches in the general population; nevertheless, little is known about their prevalence and characteristics among specific populations, particularly pharmacy and nursing students in university, as headache is widely common among undergraduate students. However, this study revealed that proper management approaches for headaches may help in preventing adverse events. Therefore, it is crucial to assess the possible management approaches for headaches among undergraduate healthcare students.

Based on the frequency of headaches, our study revealed that 66.1% reported headaches weekly, 24.2% complained of headaches five days a week, and 42.8% had four episodes of headaches in a day. A study on undergraduate students of health professions showed that 43.5% of respondents complained of headaches monthly, 37.5% had this complaint on weekly basis, and 19% experienced it daily [33]. Recently, a study in Bangladesh conducted on university students reported that more than two-thirds of the students had more than five attacks during the past month [8]. Similarly, another study among medical undergraduates at Oman University observed that 96% of the students presented at least one episode of headache during the year preceding the data collection [34]. From our study, it was found that only 34.7% of the students complained that headaches disturbed their regular activities. These results were in line with a previous study by Rafi et al. among university students, which reported that 37% of headaches caused a severe impact on their daily activity [8]. Recent evidence indicates that the progression of headaches has worsened among undergraduate students, particularly healthcare students, due to the heavy burden of assignments, laboratory activities lengthy curriculum, grading system, and increased competitive level among students, which may contribute to stress and anxiety, which consequently trigger headaches.

Only 23.7% of study participants reportedly sought medical advice. These results were in line with other previous studies published around the world [14,33,34,35]. For instance, Alkarrash et al. found that, among undergraduate students at the University of Aleppo, 15.9% sought medical attention [35]. Similarly, Panigrahi et al. reported that 10% of undergraduate students seek medical treatment for headache relief [33]. However, the percentage of students in our study who sought medical assistance for their headaches was lower than that of earlier research by Amayo et al. in a sample of medical students (50%) [36] and a sample of university students by Deleu et al. (23.3%) [34]. The variation in the medical attention in the current study compared with earlier studies might be due to discrepancies in the methodology, the definition of headache, pain intensity, study population, or real differences.

In terms of medicine use, most of the students used simple analgesics and anti-inflammatory drugs, followed by caffeine and sleeping methods to manage headaches, similar to previous findings [14,35], according to which the majority of the students used over-the-counter painkillers [14,33,35]. For example, Panigrahi et al. reported that paracetamol, aspirin, aceclofenac, and ibuprofen were the most commonly used self-medications for headaches. Additionally, students reported the use of various non-drug headache-relieving strategies such as sleeping and massaging heads [33]. These results were similar to our results. Our research revealed that headaches were typically unilateral (50.4%), bilateral (49.6%), throbbing or pulsating (61.9%), and pressing or tightening (22%) in nature, whereas a similar study reported unilateral (35.1%), bilateral (30.2%), and throbbing or pulsing (58.9%) types [33]. Additionally, according to a different study conducted among medical students, 80.9% of students had pain with a bilateral localization [37].

According to earlier studies, family history was the main reason for the incidence of headaches among participants. For instance, a previous study at Taif University, Saudi Arabia, reported that 86.6% of female students had a family history of headaches [18], while another study among medical students, by Anaya et al., revealed that 49.1% of the students had a positive family history [38]. However, our study reported that only 25.8% of the students had a family history of headaches, although it is documented that there is a 40% chance that a person would suffer from headaches if one of their parents has headache episodes. If both parents have headaches, their children have a high chance of developing them as well. In addition, there is an increased familial risk for headaches, indicating that headaches are most likely genetic. The good news is that therapies are continuously improving.

Among the general public and professionals, there is a lack of awareness about the epidemiology of headaches as well as the impact of headaches on individuals, their families, and society. Educational campaigns will create awareness about headaches, and such programs should be correctly managed in the healthcare arena. These programs will empower pharmacy and nursing students to improve headache diagnosis and management and to raise the priority of effective treatment and prevention of headaches at early stages. There is no doubt that this is a serious issue for both students and society, and additional studies are necessary to answer the risk factors of headaches.

In this study, participants’ age, gender, course of study, smoking habit, and type of headache were significantly associated with the intensity of headache, and similarly, headache intensity was associated with routine physical activity (*p* = 0.0001). Surprisingly, headache intensity was not significantly associated with academic performance (*p* > 0.005). In comparison, Panigrahi et al. found that gender, sleep disorder, consumption of soft drinks, and self-dissatisfaction with own health found to be the independent predictors of headaches. As is well known, undergraduates face difficulties in adapting to the new educational and social environment, which may lead to a stressful situation. In the case of healthcare students, this might be due to the high tension and stress subjected to them and even lack of adequate free time between courses/examinations in comparison to other non-health students.

Our study has some potential limitations. First of all, the data included herein were limited to only healthcare undergraduates (pharmacy and nursing); second, the study was conducted at a single university in Saudi Arabia, so the results may not fully reflect the headaches experienced by all healthcare undergraduates across Saudi Arabia or the general population. Third, since both male and female colleges in Saudi Arabia have separate campuses, and women are still not permitted to pursue their studies alongside men, limited responses were obtained from the female student body. Fourthly, the high number of smokers in this study may be explained by the fact that smoking is so common in modern society, especially among teenagers and college students, and that there are so many nicotine products easily available. Another potential weakness of this study is that it did not evaluate the various headache subtypes or the differences among them. Last but not least, to compare attitudes and outcomes, which can differ, this study did not include participants from a typical population of similar ages. Future studies should evaluate the generalizability of our findings by performing a more detailed assessment of different types of headaches among healthcare undergraduates. The results of this type of assessment may enable policymakers to conduct a timely assessment of societal problems and make recommendations accordingly.

## 5. Conclusions

According to the current findings, the majority of undergraduates suffered from headaches, the degree of pain was moderate, and one in every four undergraduates indicated that headaches interfered with their academic performance. Caffeine, analgesics, and anti-inflammatories were utilized to treat it. Furthermore, headache prevalence is increasing both nationally and internationally, particularly among university students, potentially leading to an increase in unfavorable effects on daily activities, including academic performance. It is critical to identify the causes generating headaches to control their occurrence. As a result, policymakers, academic officials, health professionals, and all other concerned parties should devote adequate attention to developing primary headache prevention and treatment techniques. As a result, we suggest the implementation of educational programs that teach students how to prevent and cure headache issues. This could help future graduates to overcome obstacles and live healthier lives.

## Figures and Tables

**Figure 1 medicina-59-00130-f001:**
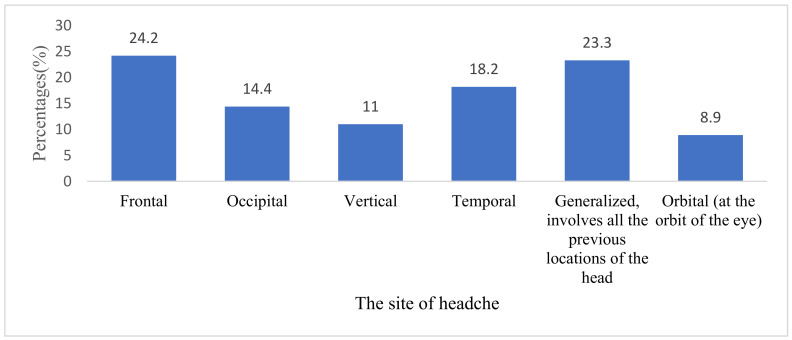
Site of headache.

**Table 1 medicina-59-00130-t001:** The demographic characteristics of the participants.

Variables	Frequency (*n*)	Percentage (%)
Gender Male Female	218 18	92.4 7.6
Age (full years) 18 to 25 26 to 30	112 124	47.5 52.5
Classification Pharmacy Nursing	124 112	52.5 47.5
Smoking status Current smoker Non-smoker Ex-smoker	134 81 21	56.8 34.3 8.9
Presence of chronic disease Yes No	40 196	16.9 83.1

**Table 2 medicina-59-00130-t002:** Characters of the headaches among respondents.

Characteristic	Frequency (*n*)	Percentage (%)
Have you ever had at least one episode of headache during the past week? Yes No	156 80	66.1 33.9
Do you have any of these fully reversible aura symptoms that are accompanied or followed by your headache episodes? Visual aura or wavy lines (Sensation disturbances) pins, needles, numbness (Speech disturbances), e.g., aphasia Motor weakness I have none of the above aura symptoms	72 10 87 51 16	30.5 4.2 36.9 21.6 6.8
Approximately, how many days have you had a headache during the past week? None One day Two days Three days Four days Five days Six days Seven days	2 47 45 42 31 57 8 4	0.8 19.9 19.1 17.8 13.1 24.2 3.4 1.7
To the previous question, how many EPISODES per day do you have a headache (Write a number) None One day Two days Three days Four days Five days Six days Seven days	2 50 31 34 101 14 1 1	0.8 21.2 13.1 14.4 42.8 5.9 0.4 0.4
Your headache is usually Bilateral Unilateral	117 119	49.6 50.4
The headache quality is usually Throbbing/Pulsating Pressing/Tightening Sharp/Stabbing	146 52 38	61.9 22.0 16.1
The average intensity of headache pain? Mean (SD) Mild pain Moderate pain Severe pain	6.72 ± 2.63 25 113 98	(M = 7) 10.6 47.9 41.5

**Table 3 medicina-59-00130-t003:** Headache impact, cause, and sensitivity among respondents.

Variables	Frequency (*n*)	Percentage (%)
Does your headache aggregate with routine activities? Yes No Not sure	82 125 29	34.7 53 12.3
Does your headache cause you to avoid routine activities? Yes No Not sure	77 143 16	32.6 60.6 6.8
Headaches associated with photophobia (sensitivity to light)? Yes No Not sure	77 151 8	32.6 64 3.4
Headaches associated with phonophobia (fear of loud sounds)? Yes No Not sure	48 181 7	20.3 76.7 3
Do you have a family history of headaches? Yes No Not sure	61 166 9	25.8 70.3 3.8
Did your headache impact your academic activity? Yes No Not sure	96 125 15	40.7 53 6.5

**Table 4 medicina-59-00130-t004:** Detailed information on the management of headaches among respondents.

Variables	Frequency (*n*)	Percentage (%)
Have you ever sought medical attention for your headache? Yes No	56 180	23.7 76.3
Which treatments/medications do you take for headaches? Analgesics and anti-inflammatories, e.g., Panadol and aspirin Sedatives Herbs and alternative and traditional medications Caffeine Message Sleeping I do not take any medications for my headaches	64 8 11 59 3 79 12	27.1 3.4 4.7 25.0 1.3 33.5 5.1
Does your headache stop when you take medication? Yes No I do not take any medications	154 48 34	65.3 20.3 14.4

**Table 5 medicina-59-00130-t005:** Cross-tabulation of undergraduate’s characters concerning intensity levels of headache.

Variables	Intensity of Headache			*p*-Value
	Mild pain 25 (10.6%)	Moderate 113 (47.9%)	Severe Pain 98 (41.5%)	
Gender Male Female	22 (10.1) 03 (16.7)	105 (48.2) 08 (44.4)	91 (41.7) 07 (38.9)	0.687
Age (full years) 18 to 25 26 to 30	22 (19.6) 03 (2.4)	75 (67) 38 (30.6)	15 (13.4) 83 (66.9)	0.0001
Classification Pharmacy Nursing	03 (2.4) 22 (19.6)	39 (31.5) 74 (66.1)	82 (66.1) 16 (14.3)	0.0001
Smoking status Current smoker Non-smoker Ex-smoker	08 (6.0) 15 (18.5) 02 (9.5)	47 (35.1) 53 (65.4) 13 (61.9)	79 (59.0) 13 (16.0) 06 (28.6)	0.0001
Presence of chronic disease Yes No	07 (17.5) 18 (9.2)	19 (47.5) 94 (48.0)	14 (35.0) 84 (42.9).	0.260
Headache type Bilateral Unilateral	17 (14.5) 08 (6.7)	61 (52.1) 52 (43.7)	39 (33.3) 59 (49.6)	0.018
Family history of headache Yes No	06 (9.4) 19 (11.0)	33 (51.6) 80 (46.5)	25 (39.1) 73 (42.4)	0.794

**Table 6 medicina-59-00130-t006:** Levels of intensity of headache concerning some activities.

Variables	Headache Intensity			*p*-Value
	Mild Pain	Moderate	Severe Pain	
Do your headaches intensify with daily activity? Yes No Not sure	06 (7.3) 19 (15.2) 0 (0)	49 (59.8) 52 (41.6) 12 (41.4)	27 (32.9) 54 (43.2) 17 (58.6)	0.008
Headache disturbs routine activity Yes No Not sure	07 (9.1) 17 (11.3) 01 (12.5)	50 (64.9) 61 (40.4) 02 (25)	20 (26.0) 73 (48.3) 5 (3.3)	0.007
Did the headache impact your academic activity? Yes No Not sure	10 (10.4) 14 (11.2) 01 (6.7)	54 (56.3) 54 (43.2) 5 (33.3)	32 (33.3) 57 (45.6) 09 (60.0)	0.182

## Data Availability

The data will be made available from the correspondence author, upon the request.

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
