# Peer review of "Assessment of Headache Characteristics, Impact, and Managing Techniques among Pharmacy and Nursing Undergraduates—An Observational Study"

_medicina, 2023, doi:10.3390/medicina59010130_

Round 1
Reviewer 1 Report
Hello Authors,
This paper follows the structure I’d expect given the title. The introduction of the content provides a relevant global picture, which is useful, especially since I don’t think there have previously been too many headache studies conducted in Saudi Arabia specifically. This research adds a different dynamic by focussing on students. This makes a lot of sense given the long hours often required to complete studies, and I’d expect particularly in these subject matters which you’ve focused on here.
Analytical tools used were Excel and SPSS which are expected in the industry. I note the majorities quoted in your sample, about the male distribution and the percentage which are smokers. You do list some potential limitations of this study, but it might be helpful to just be clear on the distributions of gender and age in this regard.
Thanks for this contribution to the field which brings the focus on undergraduate students in Saudi Arabia. Interesting dynamic and I’m sure readers would be interested to know in further studies, about what kind of headaches students tend to suffer from.
Author Response
Reviewer 1
Comments and Suggestions for Authors
Hello Authors,
This paper follows the structure I’d expect given the title. The introduction of the content provides a relevant global picture, which is useful, especially since I don’t think there have previously been too many headache studies conducted in Saudi Arabia specifically. This research adds a different dynamic by focusing on students. This makes a lot of sense given the long hours often required to complete studies, and I’d expect particularly in these subject matters which you’ve focused on here.
Response: We appreciate the comments, and we agree that headache is more prevalent among students due to various stressors and is a crucial topic among student’s
Analytical tools used were Excel and SPSS which are expected in the industry. I note the majorities quoted in your sample, about the male distribution and the percentage which are smokers. You do list some potential limitations of this study, but it might be helpful to just be clear on the distributions of gender and age in this regard.
Response: We appreciate the comments I have added it in the limitations s follows
Our study has some potential limitations. First of all, the data included herein is limited to only healthcare undergraduates (pharmacy and nursing), second, the study was conducted at a single university in Saudi Arabia, the results may not fully reflect the headaches experienced by all healthcare undergraduates across Saudi Arabia or the general population. Third, since both male and female colleges in Saudi Arabia have separate campuses and women are still not permitted to pursuits alongside men, therefore limited responses were obtained from the female students. Fourthly, the high number of smokers in this study may be explained by the fact that smoking is so common in modern society, especially among teenagers and college students, and that there are so many nicotine products easily available. Another potential weakness of this study is that it did not evaluate the various headache subtypes or the differences among them. Last but not least, to compare attitudes and outcomes, which can differ, this study did not include participants from a typical population of similar ages. Future studies should evaluate the generalizability of our findings. The results of this type of assessment may enable policymakers to assess societal problems timely and make recommendations accordingly
Thanks for this contribution to the field which brings the focus on undergraduate students in Saudi Arabia. Interesting dynamic and I’m sure readers would be interested to know in further studies, about what kind of headaches students tend to suffer from.
Response: We appreciate the comments, and we agree that headache is more prevalent among students due to various stressors and is a crucial topic among student’s
Reviewer 2 Report
As stated by the Authors, this study was aimed to assess the frequency of headaches and Headache related 18 characters and managing approaches among Pharmacy students at Saudi university, Riyadh, Saudi 19 Arabia
They reported percentages regarding not only the headache frequency but also disabling headache affecting all aspects of life and school performance as well as usage of pain killers.
I have some concerns which need to be addressed.
The authors enrolled 500 undergraduate students in PharmD and Nursing courses
A total of 236 participants completed the questionnaires. Of those 218 (92.4%) were male and 124(52.5%) were aged between 26-30, 124(52.5%) were pharmacy students, and 149 112(47.5 %) were nursing students.
The authors asserted that the aim of the study was to assess the frequency of headaches and their characteristics in these 2 under-graduated students groups. In this regard they enrolled 500 undergraduate students in PharmD and Nursing courses. What is the total number of students in the two courses from which the 500 undergraduate students are then enrolled? Which are the criteria for the enrollment? Why only a part answered to the questionnaire? Did only answer the students who suffered from headache when completing the questionnaire?
The frequency of headache obtained from the students who answered the questionnaires (236) is therefore not expressive of the frequency of headache in the total number of undergraduate students in PharmD and Nursing. Therefore the answer to this question cannot be obtained.
Furthermore half of the enrolled students chose to answer the questionnaire They only in part are representative of the entire group. This can explain the high number of males in the group of undergraduate students who answered to the questionnaire which contrast the greater prevalence in females of the most frequent primary headaches, migraine, and tension-type headache. It would be interesting to obtain information on headache characteristics with the aim to separate migraine from tension-type headache or other headaches. Do the questionnaire questions allow it?
The authors reported the frequency of attacks, and percentages of students who consider their headaches so disabling to negatively affect regular activities and academic performance but this did not allow to distinguish participants affected by migraine from those affected from tension-type headache, and to obtain percentages of low-frequency, high frequency and chronic headache forms. Nor does the percentage of patients taking painkillers or caffeine help to give a picture of the relevance of headache in the pharmacy and nursing students groups.
It is of interest to give an explanation why more pharmacy students (66.1%) than nursing students (14.3%) reported having severe headaches (p=0.0001).
I would like to suggest to modify the text (in particular methods, and results) to focus better on the aim of the study and to answer to the questions posed in the objectives (if information in this regard is available)
Author Response
Reviewer 2
Comments and Suggestions for Authors
As stated by the Authors, this study was aimed to assess the frequency of headaches and Headache related characters and managing approaches among Pharmacy students at Saudi university, Riyadh, Saudi Arabia
They reported percentages regarding not only the headache frequency but also disabling headache affecting all aspects of life and school performance as well as usage of pain killers.
I have some concerns which need to be addressed.
The authors enrolled 500 undergraduate students in PharmD and Nursing courses
A total of 236 participants completed the questionnaires. Of those 218 (92.4%) were male and 124(52.5%) were aged between 26-30, 124(52.5%) were pharmacy students, and 149 112(47.5 %) were nursing students.
The authors asserted that the aim of the study was to assess the frequency of headaches and their characteristics in these 2 under-graduated student’s groups. In this regard they enrolled 500 undergraduate students in PharmD and Nursing courses. What is the total number of students in the two courses from which the 500 undergraduate students are then enrolled? Which are the criteria for the enrollment? Why only a part answered to the questionnaire? Did only answer the students who suffered from headache when completing the questionnaire?
Response: We appreciate the comments, yes i agree that i should give more explanation about possible barriers for the recruitment. According to university counsel there were approximately 500 residential students (living in university campus) who were currently enrolled on regular basis, in both the groups. Criteria of enrollment involves, student’s living in the university hotel, and regular to the college, only Saudi nationals and headache suffers,
We followed convince sampling technique for the recruitment. Data collection as began after approaching the group leader from both the courses, most of the student’s simply avoided the survey, due to unknown reasons, some of the student’s, did not competed the survey(n=115). We sent continuous reminder and fallow ups, also we encouraged them to fill the survey, unfortunately, we received only 236 completely filled answers for the data. The eligibility was for all he student who suffers with headache
The frequency of headache obtained from the students who answered the questionnaires (236) is therefore not expressive of the frequency of headache in the total number of undergraduate students in PharmD and Nursing. Therefore, the answer to this question cannot be obtained.
Response: We appreciate the comments, yes I agree that, but it is the responses among 236 Saudi student’s from ksu
Furthermore, half of the enrolled students chose to answer the questionnaire They only in part are representative of the entire group. This can explain the high number of males in the group of undergraduate students who answered to the questionnaire which contrast the greater prevalence in females of the most frequent primary headaches, migraine, and tension-type headache. It would be interesting to obtain information on headache characteristics with the aim to separate migraine from tension-type headache or other headaches. Do the questionnaire questions allow it?
Response: Thank you for the comment, we agree with your statement however in this study we assess headache only we did not focus on separately migraine, or other types, we included all types of headache, our aim is to find the out the frequency and intensity of pain effect of it on performance we agree that females might pose more headaches types, but that may be due to their hormonal changes.
We are particularly assessing the headache which is related to stress and academic, even though we opted to include females as well due to very strict rules in Arabic culture, both are not allowed sit together, they have separate campuses, we approached female campus as well, but most of them avoided filling the survey, it may clearly indicate their carefulness or simply they avoided. since follow-up with the females is complicated and restricted.
The authors reported the frequency of attacks, and percentages of students who consider their headaches so disabling to negatively affect regular activities and academic performance but this did not allow to distinguish participants affected by migraine from those affected from tension-type headache, and to obtain percentages of low-frequency, high frequency and chronic headache forms. Nor does the percentage of patients taking painkillers or caffeine help to give a picture of the relevance of headache in the pharmacy and nursing student’s groups.
Response: We appreciate the comment, yes I agree that I should I give more explanation about different types of headache. It is not possible that all the student’s suffer with migraine, but most of the student’s during their graduation they suffer with tension type of headache, or headache due to stress, we assessed Characteristics, Impact, and Managing Techniques among student’s only, our aim is not to find the different types of headache, in future we ill like to do the study which will assess the different types of headache with multiple university student’s
It is of interest to give an explanation why more pharmacy students (66.1%) than nursing students (14.3%) reported having severe headaches (p=0.0001).
Response: It is excellent doubt asked by you, we thankfully and we appreciate, this, all of us know that PharmD course is very complicated with multiple labs and quizzes and presentations, hospital training. Furthermore, earlier study revealed that stress, sleep disturbance and reading for a longer period were the top three triggering factors of headache among pharmacy’s students comparison to nursing student’s, since nursing students have less curriculum, and duration in course m rather they have more practical training, in the case of pharmacy, student’s need to read and attend more exams, the burden high among pharmacy student’s, which makes this difference.
I would like to suggest to modify the text (in particular methods, and results) to focus better on the aim of the study and to answer to the questions posed in the objectives (if information in this regard is available)
Response: Thank you for the comment, we agree with your statement however I have the changes, as suggested and I incldued all this in the limitations. In future we would like to do the similar study with multiple university student’s, by assessing the difference types of headache as follows …
Our study has some potential limitations. First of all, the data included herein is limited to only healthcare undergraduates (pharmacy and nursing), second, the study was conducted at a single university in Saudi Arabia, the results may not fully reflect the headaches experienced by all healthcare undergraduates across Saudi Arabia or the general population. Third, since both male and female colleges in Saudi Arabia have separate campuses and women are still not permitted to pursuits alongside men, therefore limited responses were obtained from the female students. Fourthly, the high number of smokers in this study may be explained by the fact that smoking is so common in modern society, especially among teenagers and college students, and that there are so many nicotine products easily available. Another potential weakness of this study is that it did not evaluate the various headache subtypes or the differences among them. Last but not least, to compare attitudes and outcomes, which can differ, this study did not include participants from a typical population of similar ages. Future studies should evaluate the generalizability of our findings by giving more detailed assessment on different types of headaches among health care undergraduates are warranted. The results of this type of assessment may enable policymakers to assess societal problems timely and make recommendations accordingly